# DSCAM-AS1-Driven Proliferation of Breast Cancer Cells Involves Regulation of Alternative Exon Splicing and 3′-End Usage

**DOI:** 10.3390/cancers12061453

**Published:** 2020-06-03

**Authors:** Jamal Elhasnaoui, Valentina Miano, Giulio Ferrero, Elena Doria, Antonette E. Leon, Aline S. C. Fabricio, Laura Annaratone, Isabella Castellano, Anna Sapino, Michele De Bortoli

**Affiliations:** 1Center for Molecular Systems Biology, University of Turin, Orbassano, 10043 Turin, Italy; jamal.elhasnaoui@unito.it (J.E.); valentina.miano@unito.it (V.M.); giulio.ferrero@unito.it (G.F.); 2Department of Clinical and Biological Sciences, University of Turin, Orbassano, 10043 Turin, Italy; elena.doria@edu.unito.it; 3Division of Cellular and Molecular Pathology, Department of Pathology, University of Cambridge, Addenbrooke’s Hospital, Cambridge CB2 0QQ, UK; 4Department of Computer Science, University of Turin, 10149 Turin, Italy; 5Regional Center for Biomarkers, Department of Clinical Pathology, Azienda ULSS 3 Serenissima, Campo SS Giovanni e Paolo 6777, 30122 Venice, Italy; antonette.leon@aulss3.veneto.it (A.E.L.); aline.fabricio@aulss3.veneto.it (A.S.C.F.); 6Department of Medical Sciences, University of Turin, 10126 Turin, Italy; laura.annaratone@unito.it (L.A.); isabella.castellano@unito.it (I.C.); anna.sapino@unito.it (A.S.); 7Candiolo Cancer Institute, FPO-IRCCS, 10060 Candiolo, Turin, Italy

**Keywords:** lncRNA, breast cancer, alternative splicing, estrogen receptor, RNA-Seq

## Abstract

*DSCAM-AS1* is a cancer-related long noncoding RNA with higher expression levels in Luminal A, B, and HER2-positive Breast Carcinoma (BC), where its expression is strongly dependent on Estrogen Receptor Alpha (ERα). *DSCAM-AS1* expression is analyzed in 30 public datasets and, additionally, by qRT-PCR in tumors from 93 BC patients, to uncover correlations with clinical data. Moreover, the effect of DSCAM-AS1 knockdown on gene expression and alternative splicing is studied by RNA-Seq in MCF-7 cells. We confirm *DSCAM-AS1* overexpression in high grade Luminal A, B, and HER2+ BCs and find a significant correlation with disease relapse. In total, 908 genes are regulated by *DSCAM-AS1*-silencing, primarily involved in the cell cycle and inflammatory response. Noteworthily, the analysis of alternative splicing and isoform regulation reveals 2085 splicing events regulated by *DSCAM-AS1*, enriched in alternative polyadenylation sites, 3′UTR (untranslated region) shortening and exon skipping events. Finally, the *DSCAM-AS1*-interacting splicing factor heterogeneous nuclear ribonucleoprotein L (hnRNPL) is predicted as the most enriched RBP for exon skipping and 3′UTR events. The relevance of *DSCAM-AS1* overexpression in BC is confirmed by clinical data and further enhanced by its possible involvement in the regulation of RNA processing, which is emerging as one of the most important dysfunctions in cancer.

## 1. Introduction

Non-coding RNAs are an established layer of regulation in the molecular pathophysiology of complex diseases, including cancer [1]. A large amount of evidence on the oncogenic or oncosuppressor activities of non-coding RNA transcripts longer than 200 bp, defined as long noncoding RNAs (lncRNAs), has been accumulated in recent years thanks to the diffusion of deep-sequencing technologies [2,3]. In the context of Breast Carcinoma (BC), lncRNA expression and molecular activity were related to different stages of the disease as well as to the different BC subtypes [4]. Among these subtypes, the Estrogen Receptor alpha (ERα)-positive BC (Luminal A and B subtypes) represents the most frequent breast neoplasm with over 270,000 estimated new cases in the US population for 2020 [5]. Despite these tumors being characterized by a less aggressive phenotype and better patient outcome, a growing number of cases showed drug resistance and disease relapse [6]. Among the lncRNA genes related to the onset and progression of luminal BC, *HOTAIR*, *MIAT*, and *DSCAM-AS1* were described in multiple studies [7].

*DSCAM-AS1* was originally described as a lncRNA overexpressed in invasive BCs compared to normal adjacent tissue [8]. A subsequent analysis of ERα regulated lncRNAs in the luminal BC model MCF-7 performed by our group evidenced *DSCAM-AS1* as the most significantly ERα-regulated lncRNA in these cells [9]. In the same study, the *DSCAM-AS1* gene was evidenced as overexpressed in ERα-positive tumors, particularly of the luminal B subtype. Furthermore, we showed that siRNA-mediated silencing of *DSCAM-AS1* induced a reduction in MCF-7 proliferation with an increase in cell death. These results were further confirmed by different groups [10,11,12]. Niknafs and colleagues reported the overexpression of the lncRNA in tamoxifen-resistant cells whose proliferation decreases upon *DSCAM-AS1* silencing [10]. Alongside these results, our group showed that, in BC cells grown in hormone-deprived medium, ERα binding occurs in a super-enhancer region upstream of the *DSCAM-AS1* locus, which promotes the lncRNA overexpression in these cells [13]. These results, in addition to the evidence of *DSCAM-AS1* overexpression in BC patients with poor outcomes [11], make the understanding of the functional role of this lncRNA relevant.

The first evidence of interaction between *DSCAM-AS1* and RNA-binding proteins (RBPs) was reported by Niknafs and colleagues, demonstrating that *DSCAM-AS1* physically interacts with heterogeneous nuclear ribonucleoprotein L (hnRNPL) [10]. However, the functional role of this interaction was not elucidated. hnRNPL is a well-known splicing factor belonging to the heterogeneous nuclear ribonucleoprotein protein family and it is involved in the regulation of alternative splicing by binding to C/A-rich elements particularly at gene intron and 3′UTR [14,15]. Furthermore, hnRNPL activity was related to the maintenance of mRNA stability by the regulation of the nonsense-mediated mRNA decay (NMD) pathway through the binding at gene 3′UTR [16]. This activity was particularly relevant in cancer biology since the mRNA stability of well-known oncogenes like B-cell 2 (Bcl2) and Serine/arginine-rich splicing factor 3 (SRSF3), as well as the tumor suppressor protein 53 (p53), were demonstrated to be regulated by hnRNPL [16,17,18].

From a computational point of view, the recent advances in bioinformatics tools now allow the accurate characterization and quantification of gene isoforms from RNA sequencing data. These mRNA molecules, originating from the same locus, have different exon compositions and lengths and may encode for different corresponding proteins [19]. These isoforms may result from the differential usage of alternative transcription start sites (aTSSs), termination sites (aTTSs), or Alternative PolyA (APA) sites, or may be the consequence of the alternative splicing (AS) of internal exons [20]. AS is a regulatory mechanism which allows the fine-tuning of gene isoform expression in different cell types and tissues or under specific conditions such as cancer [21]. AS may affect mRNA localization, stability, and may change the open reading frame, resulting in protein isoforms with diverse functions or localization [22]. Accumulating evidence has indeed shown that different isoforms can be differentially used under distinct conditions and that this may have a substantial biological impact due to differences in their functional potentials [23,24]. These mechanisms of differential usage of isoforms, usually referred to as isoform switching, has been shown to be implicated in many diseases and is especially prominent in cancer, where it affects the expression of isoforms of genes involved in almost all cancer hallmarks [25,26].

In this study, we performed an exploratory analysis of *DSCAM-AS1* gene expression in multiple BC tumors, confirming the relation between the lncRNA expression and the poor survival rate of ERα-positive BC patients. Then, to functionally investigate the activity of this lncRNA, we analyzed the effects of *DSCAM-AS1* downregulation by RNA-seq followed by gene- and isoform-level analyses. The knockdown of *DSCAM-AS1* strongly hampers cell growth and proliferation pathways in MCF-7 cells. Interestingly, *DSCAM-AS1* knockdown not only affects gene expression but also drives remarkable changes in isoform expression and alternative splicing through its physical interaction with the splicing factor hnRNPL. Altogether, these data highlight the functional role of *DSCAM-AS1* in MCF-7 cells and shed light on the importance and diverse roles of this lncRNA in regulating isoform expression.

## 2. Results

### 2.1. DSCAM-AS1 is Overexpressed in More Aggressive ERα-Positive BCs

To investigate the expression of *DSCAM-AS1* in tumor tissues, we analyzed gene expression data from 30 public microarray datasets (Appendix A) and RNA-Seq data from The Cancer Genome Atlas (TCGA). In addition to these microarray and RNA-seq public data, we quantified, by qRT-PCR, the *DSCAM-AS1* expression in RNA samples derived from primary cancer tissues from two BC cohorts composed, respectively, of 42 (Cohort_1) and 51 (Cohort_2) subjects (details are reported in Materials and Methods and Appendix A).

Initially, we evaluated the differential *DSCAM-AS1* expression level between ER+ and ER− tumors observing, as expected, a consistent overexpression of this lncRNA in ER+ tumors (average log2FC ER+ vs. ER− = 2.07) (Figure 1a and Appendix A). This difference was statistically significant in 23 out of the 30 microarray datasets analyzed (*p*-value < 0.05), in the TCGA RNA-Seq dataset (*p*-value < 2.2 × 10^−16^) and in Cohort_1 (*p*-value = 8.01 × 10^−10^). In the TCGA dataset, the gene was detectable (fragment per kilobase per million mapped reads (FPKM) > 1) in 381 out of 803 ER+ tumors while it was detectable in 29 out of 237 ER− tumors. Noteworthily, 93% of *DSCAM-AS1* positive BCs are ER+. In coherence with this result, *DSCAM-AS1* was observed as recurrently upregulated in luminal BCs compared to non-luminal BCs (average log2FC = 2.37), with higher expression in luminal B tumors compared to luminal A BCs (average log2FC = 1.20) as previously reported [9,12]. Furthermore, in non-luminal BCs, *DSCAM-AS1* was overexpressed in HER2+ BCs (Figure 1a and Appendix A).

The association between *DSCAM-AS1* expression levels and different clinical data was instead evaluated separately for ER+ and ER− tumor groups. As reported in Figure 1a, in ER+ BCs, *DSCAM-AS1* emerged as recurrently overexpressed in high-grade BCs (G2 or G3, 10 significant datasets). In a subset of datasets, *DSCAM-AS1* was significantly overexpressed in BC with lymph node invasion (two datasets), high Ki67 levels (two datasets), HER2 overexpression (two datasets) and when diagnosed in young patients (two datasets). Finally, in four datasets, *DSCAM-AS1* was overexpressed in BCs of patients associated with a higher death rate, while in three datasets, *DSCAM-AS1* was overexpressed in BCs characterized by a higher relapse rate.

To further explore the relation between *DSCAM-AS1* expression and the patient survival rate, we performed a survival analysis using Overall Survival (OS) or Recurrence-Free Survival (RFS) data provided in nine microarray datasets and in the BRCA TCGA cohort (Appendix A). As reported in Figure 1b, the *DSCAM-AS1* overexpression was significantly associated with a higher relapse rate in GSE6532 (HR = 3.21, *p*-value = 0.0048), GSE20685 (HR = 1.9, *p*-value = 0.023) and GSE42568 (HR = 2.6, *p*-value = 0.027) datasets. In the remaining microarray datasets, patients with higher *DSCAM-AS1* expression showed a non-significant lower relapse free survival rate (Log-rank-test *p*-value range from 0.34 to 0.93), even though, in three of them (GSE88770, GSE58984, GSE19615), the separation of RFS curves was concordant (Appendix A). A nonsignificant difference (*p*-value = 0.99) was also observed in the analysis of TCGA data, as previously reported [10]. A significant difference in relapse rate was also observed among subjects from Cohort_2 (*p*-value = 0.031). With respect to the overall survival rate, *DSCAM-AS1* overexpression was significantly associated with a lower survival rate in two datasets, GSE42568 (HR = 3.47, *p*-value = 0.019) and GSE20685 (HR = 1.96, *p*-value = 0.022) (Appendix A).

### 2.2. DSCAM-AS1 Knockdown Induces a Downregulation of Cell Cycle-Related Genes in BC Cells

To investigate the functional role of *DSCAM-AS1* in ER+ BC, we performed an RNA-Seq of MCF-7 cells transfected with control or *DSCAM-AS1*-targeting LNA (locked nucleic acids) GapmeRs. The analysis of the RNA-Seq data evidenced 908 genes differentially expressed (DE, |log2FC| > 0.20, adjusted *p*-value < 0.05) upon *DSCAM-AS1* silencing (Figure 2a and Appendix A). Specifically, 420 genes showed a significant decrease in expression, while 488 genes were upregulated. Notably, the differential expression of highly significant DE genes from the RNA-Seq analysis was confirmed by qRT-PCR analysis (Figure 2b). Furthermore, we observed, in our RNA-seq dataset, the same regulation trend for four out of six genes (*BCL2*, *ESR1*, *CDC6*, *E2F7*, *FEN1*, *TOP2A*) that were previously reported to be downregulated in MCF-7 by *DSCAM-AS1* silencing using siRNA [11]. Noteworthily, *DSCAM-AS1*-silenced cells showed a lower proliferation rate compared to the control (data not shown), even though *ESR1* expression was not perturbed (Appendix A), confirming previous results [9,10]. Since the overexpression of *DSCAM-AS1* was observed not only in ERα+ but also in HER2+ breast tumors, we investigated the effects of *DSCAM-AS1* knockdown in the SK-BR-3 cell line, a commonly used HER2+ BC cell model. We observed that *DSCAM-AS1*-silencing significantly reduces the proliferation rate of SK-BR-3 cells (Appendix A), confirming that *DSCAM-AS1* is crucial for the survival of different BC cells.

A functional enrichment analysis of the DE genes showed distinct biological processes related to downregulated and upregulated genes. Specifically, downregulated genes were mainly involved in cell cycle progression, cell growth and proliferation (Figure 2c), while upregulated genes were mainly enriched in terms related to the inflammatory response, evoking type I and type II interferon signaling pathways, autophagy and the ER-phagosome pathway, apoptosis, and the regulation of cholesterol biosynthesis (Figure 2d and Appendix A).

We previously demonstrated that *DSCAM-AS1* is an ERα-regulated lncRNA [9,13]. We thus compared the list of DE genes obtained in this study with those DE upon ERα silencing [13]. Notably, 526 genes were detected as DE in both datasets (hypergeometric test *p* < 4.36 × 10^−48^, representation factor = 1.7) (Figure 2e and Appendix A). Of this, 291 genes (156 downregulated and 135 upregulated) showed coherent expression changes in the two datasets, while 235 genes showed an opposite regulation direction (Appendix A). The GO enrichment analysis of the four categories of overlapping genes showed that they are related to distinct biological processes (Appendix A). Notably, genes that showed an upregulated expression in both datasets are involved in interferon signaling pathways and the activation of the immune response, as well as autophagy regulation (Figure 2f), while genes that were downregulated in both datasets are involved in the regulation of the cell cycle process (Figure 2g). Genes upregulated only in this study but downregulated in the siRNA-ERα experiment are related to metabolic processes such as sterol and cholesterol biosynthesis (Appendix A). Finally, genes downregulated in this study but upregulated in the siRNA-ERα experiment are related to developmental processes such as gland morphogenesis and neuron projection development pathway (Appendix A).

When considering differential expression at the isoform level, the silencing of *DSCAM-AS1* perturbed the expression of 1035 isoforms, of which 439 were downregulated and 596 were upregulated (Appendix A). The DE isoforms were transcribed from 898 genes of which 381 (42%) genes were detected as DE in our gene-level analysis (Appendix A). The GO enrichment analysis of the parent genes of these DE isoforms showed enrichment in terms similar to those obtained when considering the overall gene expression changes only (Appendix A).

### 2.3. DSCAM-AS1 Physically Interacts with hnRNPL to Regulate AS in MCF-7 Cells

*DSCAM-AS1* was reported to interact with the RNA-Binding Protein (RBP) hnRNPL [10], which regulates AS and mRNA stability [15,16]. We also confirmed this interaction in our MCF-7 cellular model with an hnRNPL Cross-Linking Immunoprecipitation experiment (CLIP) (Appendix A). Therefore, in order to test whether *DSCAM-AS1* silencing displayed any effect on the AS events regulated by hnRNPL, we explored the possible regulation at the RNA isoform expression level in our dataset, as a result of either alternative exon splicing, or alternative transcription start site, or alternative poly(A) site. This was evaluated using two approaches: (i) an analysis of the isoform switching events, also called differential isoform usage, and (ii) an analysis of local AS changes occurring upon *DSCAM-AS1* silencing.

As shown in Figure 3a, using the IsoformSwitchAnalyseR tool [27], the relative contribution of the individual isoforms to the expression of the gene was evaluated by calculating an Isoform Fraction (IF) value, dividing the expression of each individual isoform by the expression level of the gene, where the latter is the sum of the expression of all of its isoforms. Next, changes in isoform fraction (dIF) upon *DSCAM-AS1* silencing were measured as dIF = IF_silencing_ − IF_control_ and significant changes were defined as isoform switching events (details are given in the Materials and Methods, Section 4.6).

As shown in Figure 3b,c, the isoform switching analysis revealed 485 genes, showing 363 significant switching events involving 578 isoforms (Appendix A). Interestingly, 254 genes (60%) have an isoform switching event with downstream consequences affecting the functional properties of 320 isoforms (Appendix A). Notably, the annotation of the isoforms upregulated upon *DSCAM-AS1* silencing revealed the enrichment of 5′UTR lengthening and 3′UTR-shortening events (Figure 3d). Equally, the analysis revealed that the switching events involved AF and AL prevalently compared to multiple or single exon skipping events (Figure 3e). The switching genes were involved in distinct pathways, including the gland development pathway (*PTPN3*, *RTN4*), cell cycle progression pathway (*POLA1*, *MYCBP*), positive regulation of cell growth (*ZFYVE27*, *TGFBR1*, *TNFRSF12A*, *RPTOR*), apoptosis pathway (*FEM1B*, *BCL2*), and pre-mRNA splicing pathway (*SRSF5*). Interestingly, the *BCL2* gene, whose RNA stability was previously demonstrated to be regulated by hnRNPL [16], was characterized by 3′UTR shortening upon *DSCAM-AS1* downregulation (Figure 3f). The list of switching events affecting the 3′UTR length is reported in Appendix A. An analysis of alternative 3′UTR usage from TCGA BC-data revealed a significant relationship between *DSCAM-AS1* expression and the alternative use of the 3′UTR of 360 genes (Appendix A). Among them, 27 were also detected in our analysis in MCF-7 cells (Figure 4a), including the shorter BCL2 3′UTR, whose usage was negatively correlated with the *DSCAM-AS1* expression (Figure 4b,c), in coherence with our isoform-switching analysis.

In addition to the isoform switching analysis, which detects occurring changes by considering the whole transcript sequence, we performed an analysis of changes involving local AS events upon *DSCAM-AS1* silencing using the Whippet tool [28], which quantifies the inclusion levels for seven AS event types (Figure 5a). The analysis identified 2085 significant AS events (Posterior probability *p* > 0.9 and |ΔPSI| > 0.1), differentially regulated upon *DSCAM-AS1* silencing, affecting the splicing pattern of 1339 genes (Figure 5b and Appendix A). Interestingly, the most frequent AS event in our data was tandem Alternative PolyA (APA) sites (1326 events), resulting in the shortening of isoforms from 497 genes (Figure 5b). The biological processes involving the genes affected by AS events were mainly related to cell cycle progression, centrosome regulation, the regulation of metabolism, and ncRNA processing (Figure 5c and Appendix A). Interestingly, 48 out of 76 genes related to the cell cycle progression process had an APA event, followed by 15 genes with different alternative Transcriptional Start Sites (TS).

### 2.4. hnRNPL-Binding Motif is Enriched Around Sites of AS Events Regulated by DSCAM-AS1 Silencing

To identify putative RNA-binding proteins (RBPs) regulating the observed AS changes upon *DSCAM-AS1* silencing, we performed an RBP-binding motif enrichment analysis within the sequences of the exons and the flanking regions involved in AS events. For cassette exon skipping events, the analysis revealed a significant enrichment of the hnRNPL-binding motif (Figure 5d, left panel) within the 200 nucleotides upstream and downstream of the skipped exon. No enrichment was observed within exons, in agreement with the known intronic binding of hnRNPL [29]. This enrichment was observed only for exons characterized by a lower inclusion level upon *DSCAM-AS1* silencing. In contrast, a different subset of RBPs was identified to be enriched for exons showing more inclusion levels upon *DSCAM-AS1* silencing (Figure 5d, right panel). Interestingly, the enrichment analysis revealed a significantly higher presence of the hnRNPL-binding motif in 3′UTR-shortening than in 3′UTR-lengthening events when considering the gene region involved in APA events (Chi-squared test *p*-value = 5.716 × 10^−08^) (Figure 5e). Among the 3′UTR-shortening events, the hnRNPL motif was observed to be enriched in regions upstream both of the proximal or of the distal polyA site (Figure 5f,g). The list of RBPs predicted to be enriched for the 3′UTR-lengthening events is shown in Appendix A and the full list of enriched RBPs enriched for different AS events is reported in Appendix A.

## 3. Discussion

In this study, by analyzing *DSCAM-AS1* expression levels across multiple cohorts of BC patients’ samples, we confirm *DSCAM-AS1* overexpression in ER+ tumors and its correlation with a worse prognosis. We clearly demonstrate the association of this overexpression with undifferentiated and more aggressive tumors. Furthermore, we show, in MCF-7 BC cells, that *DSCAM-AS1* silencing strongly impairs gene expression. More importantly, we show for the first time that *DSCAM-AS1* influences AS and the alternative 5′ and 3′ ends of mRNA transcripts, possibly through its interaction with hnRNPL. Since alteration of splicing patterns is emerging as an important hallmark of cancer, we believe that *DSCAM-AS1* may represent an important target for bringing to light novel aspects of BC biology.

The analysis of *DSCAM-AS1* expression in BC samples confirmed the prevalent expression of this lncRNA in ERα-positive BCs of the luminal subtype, as previously shown [9,10]. In coherence with the results from Niknafs and colleagues [10], in the TCGA data, *DSCAM-AS1* expression was not significantly related to particular clinical features or to a different patient outcome. Conversely, our meta-analysis of microarray data clearly highlights the *DSCAM-AS1* overexpression in more aggressive and less differentiated ER-positive BCs. Indeed, *DSCAM-AS1* expression was observed to be increasingly expressed from well to less differentiated BCs and, noteworthy, *DSCAM-AS1* expression was positively related to disease relapse in three datasets, as previously reported in an independent BC cohort from the study of Sun and colleagues [11]. These results are consistent with the previous observation of an increase in *DSCAM-AS1* expression in drug-resistant BCs and cell lines, including those resistant to Tamoxifen, a selective ERα modulator [10,30]. Interestingly, in a recent single-cell analysis of MCF-7 cells resistant to aromatase inhibitors, *DSCAM-AS1* emerged as increasingly expressed in a population of drug-resistant cells [31]. However, despite these promising results, clear evidence of the role of *DSCAM-AS1* as a reliable prognostic marker of BC relapse or drug resistance requires further investigation in a dedicated prospective study. It is interesting to note that *DSCAM-AS1* was also detected as highly expressed in HER2-amplified, ER-negative tumors, suggesting an ERα-independent mechanism enhancing its expression in this BC subtype. As previously shown by our group, a super-enhancer region is localized upstream of the *DSCAM-AS1* locus and it is involved in the regulation of this lncRNA [13]. This genomic region can represent a platform for the recruitment of different transcriptional regulators, including the Transcription Factor AP2-γ (AP2-γ), which was previously reported to lead overexpression of the *ERBB2* gene [32] and, in complex with other transcription factors, it could drive the *DSCAM-AS1* overexpression.

Silencing *DSCAM-AS1* by LNA GapmeRs transfection deeply altered the expression of hundreds of genes in MCF-7 cells. We chose LNA GapmeRs since, in comparison with siRNAs, they were demonstrated to be more stable, more efficient in targeting nuclear RNAs, and they rely on the activation of RNaseH, which cleaves the RNA:LNA hybrids, avoiding the saturation of silencing machineries, like the siRNA/AGO complex [33]. Strikingly, by comparing the gene expression changes occurring upon *DSCAM-AS1* silencing to those occurring upon ERα silencing [34], both *DSCAM-AS1* and ERα were predicted to be involved in the same pathway regulating cell growth and survival of MCF-7 BC cells, consistent with the role of *DSCAM-AS1* as a downstream effector of the ERα signaling pathway [13]. These results are in line with those from Sun and colleagues, reporting that siRNA-mediated *DSCAM-AS1* silencing induces cell cycle arrest in MCF-7 cells [11]. In addition, another study by Xu et al. confirmed the elevated expression of *DSCAM-AS1* in BC cells and showed that the silencing of *DSCAM-AS1* inhibited proliferation and cycle progression, as well as increased cell apoptosis in vitro [35]. Finally, overexpression of *DSCAM-AS1* in T-47D and ZR-75.1 BC cell lines, which show lower expression level of the lncRNA compared to MCF-7 cells, confers a proliferative advantage and a pronounced migratory phenotype [10], suggesting a *DSCAM-AS1* oncogenic role in BC progression.

The expression of many cell cycle-related genes was strongly hampered by *DSCAM-AS1* silencing, including MYC proto-oncogene, bHLH transcription factor (*MYC*), a key regulator of cell growth, proliferation, and apoptosis [36], ret proto-oncogene (*RET*), a well-known proto-oncogene which regulates cell proliferation and survival and a direct target of the ERα signaling pathway [37], Topoisomerase II alpha (*TOP2A*), a proliferation marker whose higher expression in BC is associated with higher tumor grade and the Ki67 index [38]. Furthermore, knocking down *DSCAM-AS1* also reduced the expression levels of genes involved in DNA replication, such as *POL2A,* as well as genes involved in DNA unwinding processes such as mini-chromosome maintenance complex component genes, *MCM4* and *MCM7*. Moreover, the RNA-seq analysis revealed a significant downregulation of apoptosis-related genes such as *BCL2*, an anti-apoptotic gene which has been previously reported to be downregulated upon knocking down *DSCAM-AS1* with siRNA [11].

Given the reported interaction between *DSCAM-AS1* and the splicing factor hnRNPL, also confirmed in this study, we analyzed the expression changes upon *DSCAM-AS1* silencing at isoform level, since these important events are undetectable when the analysis is related only to gene level. Indeed, the expression analysis at the isoform level reveals that almost more than half of the genes with at least one DE isoform in our RNA-seq dataset were not likewise classified as DE when considering gene level measurements only (Appendix A). Although a direct GO analysis of isoforms is not applicable, we found that the genes with DE isoforms are involved in cellular processes similar to those observed for DE genes that were identified by gene-level analysis. This result demonstrates that, by different mechanisms, at both gene and isoform levels, *DSCAM-AS1* silencing affects the same pathways in MCF-7 cells. Furthermore, we applied an isoform switching analysis to identify cases where DE isoforms of the same gene were dysregulated in opposite directions, indicating a change in the expression of the gene that could not be appreciated by a classical gene-level analysis. Thus, we believe and foresee that genes showing isoforms dysregulated in opposite directions could be studied further to decipher their associated relevance with the levels of expression of *DSCAM-AS1* in the context of BC.

Interestingly, the RNA-binding motif enrichment analysis we performed revealed distinct subsets of RBPs as putative regulators of AS events. Noteworthily, all the RBPs predicted to be enriched are highly expressed in MCF-7 cells and five of them (CELF1, HNRNPA1, RBM24, MBNL3, SFPQ) were differentially expressed or spliced upon *DSCAM-AS1* silencing (Appendix A). Specifically, the hnRNPL-binding motif was predicted as the most frequent, among other RBPs motifs, in the case of exon skipping events, with almost all the predicted binding sites located in intronic regions upstream or downstream the regulated cassette exon. This is in line with previous evidence from literature showing that hnRNPL inhibits exon inclusion through its preferential binding to CA-rich splicing silencer elements located either upstream or downstream of the regulated exons [29]. The intronic binding motif enrichment of hnRNPL suggests that downregulation of *DSCAM-AS1* may reduce the fraction of hnRNPL engaged in the interaction, increasing the fraction available to regulate cassette exon splicing in other target RNAs. However, a more complex positional code has been proposed by recent studies to explain the activating or inhibiting activity of hnRNPL on target exons [14]. Thus, we believe that further experiments are strictly needed to decipher the mechanism or mechanisms by which *DSCAM-AS1* regulates the function or the binding of hnRNPL. We also found other RBPs known to inhibit exon inclusion such as HNRNPA1, a member of the heterogeneous nuclear ribonucleoprotein protein family [39]. In contrast, the binding motifs of a different set of RBPs, known to induce exon inclusion, such as CUG-Repeat Binding Protein (CUGBP) Elav-like family member 1 (CELF) and Splicing Regulator (SR) proteins, were predicted to be enriched in the case of exon inclusion events. Interestingly, among the enriched RBPs identified around *DSCAM-AS1*-mediated splicing events, we found PCBP2, which was previously reported to bind *DSCAM-AS1* in an RNA-pulldown coupled to a mass spectrometry experiment [10]. PCBP2 is a protein known to interact with hnRNPL [40], suggesting a complex network of splicing regulation related to *DSCAM-AS1*-hnRNPL interaction.

Furthermore, in addition to hnRNPL, we successfully identified a set of RBPs previously known to regulate APA site selection in the case of 3′UTR-shortening/lengthening events, including muscleblind-like proteins 1, 2 and 3 (MBNL1, MBNL2 and MBNL3), which are known to bind preferentially to 3′UTR regions [41], Ras guanosine triphosphate (GTP)-ase-activating protein-binding protein 1 (G3BP1), an essential splicing factor for normal stress granule assembly and, consequently, the preservation of polyadenylated mRNAs [42], Splicing Factor proline/glutamine rich (SFPQ) known to facilitate miRNA-target binding [42,43], as well as the RNA-binding motif 47 protein (RBM47), which was previously reported to inhibit the proliferation of different BC cell lines and whose binding was predominantly at 3′UTRs [44].

*DSCAM-AS1* silencing had a strong effect on alternative polyadenylation site selection, in accordance with the 3′UTR-shortening events identified by the isoform switching analysis. 3′UTR shortening driven by alternative polyadenylation was previously shown to be extensively present both in normal and in cancer cells [45] and, in the latter, it could involve both oncogenes and tumor suppressor genes [46,47,48]. Our findings of 3′UTR-shortening events are in line with a study by Wang and colleagues, where they show the enrichment of shortened 3′UTRs in samples derived from BC patients and characterized by a low proliferation rate [49]. Interestingly, in our data, 24 different genes are characterized by 3′UTR shortening upon *DSCAM-AS1* silencing and are annotated to cell cycle, DNA replication or apoptosis-related terms, including *BCL2*, *CASP2*, *CDKN2C*, *EGF, CEP290*, and *CCNE1* (Appendix A). The stability of the antiapoptotic gene *BCL2* was previously reported to be regulated by hnRNPL, which prevents the trigger of the NMD by directly interacting with the longer 3′UTR of the gene [16]. However, the study from Lim and colleagues suggested that the 3′UTR splicing of *BCL2* is independent of hnRNPL in MCF-7, proposing the presence of an unknown factor interacting with the hnRNPL in the regulation of this splicing event [50]. In our data, the shortening of the *BCL2* isoform, as well as the gene downregulation, is coherent with the impairment of hnRNPL-mediated stability, which could be directly mediated by the interaction with *DSCAM-AS1*. Furthermore, the analysis of data of 3′UTR usage in tumor samples from TCGA confirmed differential *BCL2* 3′UTR usage in relation to the *DSCAM-AS1* expression. Despite the interaction between *DSCAM-AS1* and hnRNPL at BCL2, 3′UTR deserves further experimental validations, and *DSCAM-AS1* represents a good candidate to regulate this process in luminal breast cancer cells.

Furthermore, in the study of Xue and colleagues, *CCNE1* 3′UTR lengthening was observed to be recurrently detected in six cancer types including BC [46]. However, to our knowledge, no evidence of the interaction between *CCNE1* 3′UTR and hnRNPL was previously described, and definitely, further investigations are needed to clarify the functional consequence of the 3′UTR alteration on the genes identified in our analysis. Indeed, the prediction of the functional consequence of a 3′UTR-shortening event is not trivial and requires dedicated experimental validation. This is due to the heterogeneous and connected molecular pathways that can be affected by such an AS event, including a widespread alteration in the network of microRNA–target interactions [48], inhibition or induction of the NMD pathway [51], or the generation of novel RNA fragments [52].

The hypothesis of a *DSCAM-AS1*-mediated regulation of hnRNPL interaction at gene 3′UTR is supported by previous evidence of lncRNA–RBP interaction in the post-transcriptional regulation of mRNAs, including the linc-RoR mediated increase in the mRNA stability of c-Myc gene by interaction with hnRNPI and AUF1 at gene 3′UTR [53]. Noteworthily, while hnRNPL expression was not affected upon *DSCAM-AS1* silencing (Appendix A), the hnRNPL-binding motif was among the most enriched in the case of exon skipping and alternative polyadenylation site events, suggesting that the observed AS changes might be caused by the disruption of the hnRNPL–*DSCAM-AS1* interaction upon lncRNA silencing. Our hypothesis about the direct activity of hnRNPL on the detected AS events upon *DSCAM-AS1*-silencing was further supported by data analysis of published studies on hnRNPL activity and RNA-binding profile in different cell lines [14,15,54]. Specifically, 47 out of 119 cassette exon events occurring upon LNA-mediated *DSCAM-AS1* silencing were observed also upon siRNA-mediated silencing of hnRNPL in LNCaP, an androgen receptor-positive prostate cancer cell line [15] (Appendix A). Furthermore, by overlapping the coordinates of the genomic regions involved in the AS events with data of hnRNPL–RNA interactions from CLIP/RIP (RNA ImmunoPrecipitation) -based assays from these three studies, we observed 605 out of 2085 AS events identified in our analysis characterized by hnRNPL-binding events detected in at least one study (Appendix A). These results support a direct role of hnRNPL on the regulation of the AS pattern of these genes.

Finally, we fully exploited our RNA-Seq dataset potential and analyzed changes occurring upon *DSCAM-AS1* silencing at both gene and isoform levels. Importantly, by computational analysis, we successfully identified changes in AS, further supporting our *DSCAM-AS1*-hnRNPL interaction hypothesis. Further experiments, such as hnRNPL CLIP-seq, could shed light on the precise molecular mechanisms by which *DSCAM-AS1* interacts with hnRNPL to regulate alternative splicing.

## 4. Materials and Methods

### 4.1. Experimental Part

#### 4.1.1. Cell Culture and LNA GapmeR^™^ Transfection

MCF-7 and SK-BR-3 cells were routinely grown in DMEM (Dulbecco’s Modified Eagle’s Medium) (Life Technologies, 31053–028, Waltham, MA, USA) supplemented with 10% heat-inactivated FBS (fetal bovine serum) (Euroclone S.p.A, ECS0180L, Milan, Italy) and 2 mM L-glutamine (ThermoFisher Scientific, 25030–024, Waltham, MA, USA). Batches of human cell lines were purchased from ATCC (American Type Culture Collection). Cells were cultured at 37 °C with 5% CO2. The cell transfection in suspension was performed by seeding one million MCF-7 or SK-BR-3 cells along with a transfection mix composed of LNAs (20 nM final concentration) and Lipofectamine3000 (ThermoFisher Scientific, L3000015, Waltham, MA, USA), as transfecting reagents. Cells were left to attach overnight in the cell incubator and, subsequently, the medium was refreshed. Experiments were performed in three biological replicates 48 h after LNA transfection. Three custom-designed LNAs GapmeR^™^ from Exiqon were used to target all the DSCAM-AS1 lncRNA isoforms (LNA_1 5′-ATGGCAGTTGGAGGAG-3′, LNA_2 5′-ACAGAGAAGGACATGG-3′ and LNA_3 5′-AAGTAGCTTCATCTTT-3′); negative control A-LNA longRNA GapmeR™ was used as a control LNA (5′-AACACGTCTATACGC-3′); Exiqon, 300611-00. Now Exiqon is part of Qiagen company, Hilden, Germany.

#### 4.1.2. RNA Isolation and Quantitative Real Time PCR (qRT-PCR) Analysis

Total RNA used for downstream qRT-PCR analysis was isolated from MCF-7 and SK-BR-3 cells using the PureZOL™ reagent (BioRad, 7326890, Hercules, CA, USA), according to the manufacturer’s protocol. Total RNA used for downstream RNA-seq analysis was isolated from MCF-7 cells using the RNeasy Mini Kit, according to the manufacturer’s protocol (Qiagen, 74104, Hilden, Germany). Frozen breast cancer tissues of Cohort_1 [9] and breast cancer tissue powders of Cohort_2 (supercooled biopsies pulverized using a micro-dismembrator (Braun, Melsungen, Germany) and stored at −80 °C) were directly homogenized in PureZOL™ to extract total RNA. All total RNA samples purified with PureZOL™ reagent were subjected to DNase treatment to remove contaminating genomic DNA (Invitrogen™ ezDNase™ Enzyme, ThermoFisher Scientific, 11766051). First-strand cDNA synthesis was performed with the SensiFAST™ cDNA Synthesis Kit (Bioline, BIO-65054, London, UK). qRT-PCR analysis was performed using the SYBR^®^-green method (SensiFAST SYBR^®^ Hi-ROX Kit, Bioline, BIO-92005, London, UK). Real-time PCR primers were custom-designed or purchased from Qiagen (249900 QuantiTect Primer Assay, Qiagen, Hilden, Germany) and are listed in Appendix A. The expression of *18S* was used to normalize the expression level of specific targets.

#### 4.1.3. Quantification of DSCAM-AS1 Expression in Primary Tumor Tissue Samples

The *DSCAM-AS1* expression was measured as described above in primary tissue samples from two different patient cohorts, namely Cohort_1 and Cohort_2.

Cohort_1 consists of 42 samples and these were analyzed as described in [9] with updated follow-up information. The study was conducted in accordance with The Code of Ethics of the World Medical Association (Declaration of Helsinki) for experiments involving humans. The research protocol was approved by the Research Ethics Committee of the University of Turin. Considering that the present retrospective study did not modify the patients’ treatment and was conducted anonymously, no specific written informed consent was required.

Cohort_2 consists of 51 samples deriving from the Certified Biobank of the Regional Center for Biomarkers, Department of Clinical Pathology, Azienda ULSS 3 Serenissima, Venice, Italy. The samples were obtained from patients with primary BC previously collected for diagnostic purposes, whose data were anonymized by coding. For these samples, archived for more than twenty years, the need for informed consent was waived by the ethics committee in accordance with the requirements of Italian law (Italian Data Protection Authority—Garante Privacy, Authorization no. 9/2014—General Authorization to Process Personal Data for Scientific Research Purposes. Published in Italy’s Official Journal No. 301 on 30th December 2014).

In Cohort_2, *DSCAM-AS1* expression levels were considered high (1 = positive) if values of dCT from qRT-PCR analysis were <16 or >16 when *DSCAM-AS1* CT value was <28. Conversely, *DSCAM-AS1* expression levels were considered low (0 = negative) if values of dCT from qRT-PCR analysis were >16 or <16 when *DSCAM-AS1* CT value was >28. The complete clinical information of the subjects included in these cohorts is reported in Appendix A. Differences in *DSCAM-AS1* levels measured in samples from these subjects were statistically evaluated using the Wilcoxon rank-sum test and Chi-square test.

#### 4.1.4. Cross-Linking and Immunoprecipitation (CLIP) of hnRNPL

In total, 2.5 million MCF-7 cells were seeded in 15cm diameter dishes and cultured for 48 h before the crosslinking with 0.37% Formaldehyde (Merck Millipore, F8775, Burlington, MA, USA) for 10 min at 37 °C. The crosslinking reaction was stopped with 125 mM Glycine (Biorad, 1610717, Hercules, CA, USA) for 5 min at room temperature. Cells were then washed twice in 1X PBS supplemented with 1% protease inhibitor cocktail (Merck Millipore, P2714, Burlington, MA, USA) and PMSF (Phenylmethylsulfonyl Fluoride) (Merck Millipore, 93482, Burlington, MA, USA), collected by scraping in 1ml of 1X PBS, pelleted by 5 min centrifugation 4000 rpm at 4 °C and lyzed for 30 min on ice with RIPA lysis buffer (50 mM Tris HCl, 150 mM NaCl, 1.0% NP-40, 0.5% Sodium Deoxycholate, 1.0 mM EDTA, 0.1% SDS) supplemented with 1% protease inhibitor cocktail (Merck Millipore, P2714, Burlington, MA, USA), PMSF (Merck Millipore, 93482, Burlington, MA, USA) and RNase-inhibitor (RNaseOUT™ Recombinant Ribonuclease Inhibitor, ThermoFisher Scientific, 10777019, Waltham, MA, USA). Total extracts were sonicated for three cycles (20″ ON, 30″ OFF) by using an immersion sonicator device. After spinning for 10 min, 13,000 rpm at 4 °C, MCF-7 extracts were incubated for 2 h with 30 µL of BSA-coated beads (Protein G Sepharose 4 Fast Flow, GE Healthcare 17061801, Chicago, IL, USA) for sample pre-clearing. The 5% of total volume was collected as an input and stored at −20 °C until the end of the immuno-precipitation steps. Pre-cleared samples were incubated O/N at 4 °C with 10 µg of normal rabbit IgG (Merck Millipore, 12-370, Burlington, MA, USA) or of two different anti-hnRNPL antibodies (anti-hnRNPL 4D11 and D-5, Santa Cruz Biotechnology, sc-32317 and sc-48391, Dallas, TX, USA) on a rotating platform. IP (immunoprecipitation) samples were then incubated for 2 h with 50 µL of BSA-coated beads (Protein G Sepharose 4 Fast Flow, GE Healthcare 17061801, Chicago, IL, USA). IP samples were then washed five times in the RIPA lysis buffer. After the washing steps, dried beads and input samples were resuspended in RIPA lysis buffer without SDS and de-crosslinked at 65 °C for 1 h at 650 rpm by adding Proteinase K (Proteinase K Solution 20 mg/mL, ThermoFisher Scientific, AM2546, Waltham, MA, USA). Immuno-precipitated RNA was extracted with PureZOL™ reagent (BioRad, 7326890, Hercules, CA, USA), according to the manufacturer’s protocol, cDNA was synthetized and analyzed by qRT-PCR as described above. hnRNPL-CLIP enrichment was normalized on input samples and expressed as the enrichment of specific binding over nonspecific IgG control binding.

#### 4.1.5. Cell Proliferation Assay by Crystal-Violet Staining

SK-BR-3 cells were transfected with LNA–CTRL or LNA–DSCAM-AS1 as described and 5 × 10^5^ cells/well were seeded in a 96-MW microtiter plate in five technical replicates for each experimental condition. After 48 h from transfection, the culture medium was removed, and cells were washed twice in 1X PBS and fixed with methanol for 5 min. The staining was carried out with a 1% Crystal Violet solution for 10 min at room temperature (Merck Millipore, V5265, Burlington, MA, USA). Several washes with dH_2_O were performed to remove excess staining. The incorporated dye was solubilized and eluted with 10% glacial acetic acid solution and the absorbance was measured at 595 nm.

#### 4.1.6. Western Blot

Whole-cell lysate was harvested in boiling lysis buffer (25 mM Tris·HCl pH 7.6, 1% SDS, 1 mM EDTA, 1 mM EGTA) and 50 µg of total protein extract was loaded into an 8% Acrylamide gel. The antibodies used were designed against hnRNPL (Santa Cruz Biotechnology, sc-48391, Dallas, TX, USA) and GAPDH (Santa Cruz Biotechnology, sc-32233, Dallas, TX, USA). Densitometry readings were measured by the Image Lab Software (Biorad, version number 6.0.1.) and intensity ratio calculations are reported in Appendix A. Uncropped western blot images are reported in Appendix A.

### 4.2. Analysis of DSCAM-AS1 Expression with Respect to Different Clinical Data

The analysis of the *DSCAM-AS1* in tumors characterized by different clinical parameters was performed by considering the data from 30 public microarray experiments (Appendix A) and from TCGA. Only microarray datasets characterized by at least 30 samples were selected. Furthermore, since *DSCAM-AS1* is a well-known ER-alpha target, only datasets composed of ER+ tumor subtypes or associated with the information of the ER status were selected in order to separately perform the analysis on ER+ and ER− tumors. The analysis of the *DSCAM-AS1* expression with respect to different clinical data was performed using shinyGEO tools [55] in default settings. The analysis was performed only for clinical data associated with at least three samples per class analyzed. shinyGEO was also used to perform the survival analyses on datasets associated with information on patient overall survival or relapse-free survival.

The analysis of DSCAM-AS1 expression in TCGA datasets was performed by retrieving the processed RNA-Seq datasets from the GDC (Genomic Data Commons) data portal [56]. Specifically, the FPKM expression levels of 1040 subjects were retrieved from this portal. DSCAM-AS1 expression was evaluated with respect to subjects’ clinical covariates. Clinical data were obtained from cBioPortal [57], considering the clinical data from the dataset named “Breast Invasive Carcinoma (TCGA, Firehose Legacy)”. The analysis was performed separately for ER+ and ER− tumors separated based on the IHC level of ER. The significance of the differential expression between two sample classes was computed using the Wilcoxon rank-sum test. Survival analysis on TCGA data was performed using the survival v3.1 R package and ggsurvplot function of the survminer 0.4.6 R package.

### 4.3. RNA Sequencing Libraries Preparation and Data Analysis

MCF-7 cells were transfected with control or *DSCAM-AS1*-targeting LNAs and RNA was extracted 48 h after transfection, as described above. An RNA quality check (RNA integrity number (RIN) > 8) was achieved with a Fragment Analyzer (Advanced Analytical Technologies, Inc., Ankeny, IA, USA) and quantified with Qubit (Qubit™ RNA HS Assay Kit, ThermoFisher Scientific, Waltham, MA, USA; Q32852). Two µg of RNA were polyA+ selected and RNA-Seq libraries were constructed using Illumina TruSeq RNA sample prep kit (TruSeq™ RNA Sample Prep Kit v2-Set B, Illumina, RS-122-2002, San Diego, CA, USA). Paired-end (PE) cluster generation was performed using cBot on Flow Cell v3 (TruSeq PE Cluster Kit v3-cBot-HS, Illumina, PE-401-3001, San Diego, CA, USA). The sequencing of libraries was performed on the HiSeq sequencing system (Illumina, San Diego, CA, USA). The raw RNA-Seq data were deposited Gene Expression Omnibus (GEO) with the identifier GSE150591.

A flow chart of the mRNA sequencing data analysis, including pre-processing and quality control, quantification at both gene and isoform levels, the identification of differentially expressed genes and isoforms and the identification of differentially regulated alternative splicing (AS) events is summarized in Appendix A.

Raw reads were assessed for Phred quality scores using the FASTQC program (https://www.bioinformatics.babraham.ac.uk/projects/fastqc/), and low bases and adaptor sequences were trimmed off using Fqtrim (http://ccb.jhu.edu/software/fqtrim/), retaining only reads of 75 bass length. Then, clean reads were aligned against the human reference genome (GRCh38.p10 assembly) with Gencode v27 annotation (gencode.v27.annotation.gtf.gz) using STAR v2.5.1b [58]. STAR was run in two-pass mode, allowing alignment to the transcriptome coordinates by setting the option quantMode to TranscriptomeSAM; summary statistics of read alignment per sample are given in Appendix A. Next, the expression levels in read counts, transcript per million fragments mapped (TPM), and FPKM units were then estimated at both gene and isoform levels by running RSEM (RNA-Seq by Expectation Maximization) [59] on the alignment files in default parameters.

### 4.4. Differential Expression Analysis

Differentially expressed genes and isoforms upon DSCAM-AS1 silencing compared to the control condition were identified using the DESeq2 R package (v1.26.0) in default parameters [60]. The expression at isoform level was summarized to gene level using the *tx-import* bioconductor package [61] and the resulting count matrices were provided to DESeq2. Prior to DE analysis, lowly expressed genes and isoforms were discarded from the analysis and only genes or isoforms with more than 10 normalized read counts in at least one condition (three out of six samples) were considered for further downstream analysis. A gene or isoform was assigned as differentially expressed if its associated BH-adjusted *p*-value < 0.05. For all the data visualization plots, including heat maps and volcano plots, MA (mean average) plots were made using the ggplot2 R package (v.3.2.1) [62]. A quality control check of the replicates is given in Appendix A.

### 4.5. Gene Ontology Enrichment Analysis

Gene ontology terms enriched for upregulated and downregulated genes were obtained using the Gene Annotation and Analysis Resource Metascape program [63]. The list of upregulated and downregulated genes were analyzed separately using the Single List Analysis option. The statistically enriched GO terms related to each category of genes were obtained from the GO Biological Processes branch. Only GO terms that were associated with an enrichment factor > 1.5 and an accumulative hypergeometric test with an adjusted *p*-value < 0.05 were considered to be significant. To reduce redundancy, the GO terms showing a high number of overlapping genes and a large degree of redundancies were clustered into groups based on their degree of similarity and each group or cluster was represented by the most significant GO term. The top 20 significant clusters were selected for visualization purposes.

### 4.6. Isoform Switching Analysis

To test for isoform switching events, IsoformSwitchAnalyzeR tool was applied [27]. Briefly, from the RNA-seq data, the tool takes, as inputs, isoform expression levels quantified in transcript per million fragments mapped (TPM) units normalized to transcript length and then calculates an isoform fraction (IF) ratio by dividing the isoform expression with the expression of the parent gene (TPM_iso_/TPM_gene_). Lowly expressed genes with less than 1 TPM and lowly expressed isoforms not contributing to the expression of the gene (IF < 0.01) were excluded from downstream analysis. The IF was then calculated per each of the remaining isoforms and per condition. For each isoform, a dIF (IF_silencing_ − IF_control_) representing the difference in isoform usage between the two conditions was calculated. A cut-off criterion was applied by selecting only those isoforms for which DSCAM-AS1 silencing induced a significant change (BH-corrected *p*-value ≤ 0.05) in IF by at least 10% (i.e., |dIF| > 0.1). Next, the sequences corresponding to those isoforms showing significant switching events upon DSCAM-AS1 silencing were extracted and then annotated for the presence of signal peptide sequences, coding potential and for their associated pfam protein domains using signalP [64], CPC2 [65] and Pfam [66] tools, respectively. The biological consequences of the observed switches, including intron retention, domain gain/loss, coding/non-coding potential and shortening/lengthening of the open reading frame were then evaluated for the switching isoforms from the same parent gene. Next, according to the applied annotation on the switching isoforms, genes were classified into genes with or without downstream functional consequences.

### 4.7. Differential Alternative Splicing Analysis

The list of differentially regulated AS events upon *DSCAM-AS1* silencing were identified using Whippet [28]. Whippet is a tool which takes, as inputs, a gene annotation model file together with a genome file and generates a contiguous splice graph index representation of each gene included in the annotation file. The reads are then directly mapped against the contiguous splice index. Splicing events are represented as nodes by Whippet, where each node corresponds to an exonic region of the gene and the incoming and outgoing edges to each node define the set of reads supporting its inclusion and exclusion, respectively. Whippet provides a splice index (PSI) as a measure of the inclusion level of each node by calculating the ratio of the paths supporting the inclusion of the node divided by the total number of the paths supporting both the inclusion and exclusion of that node. Whippet quantifies changes in different possible splicing events including Alternative First (AF) or Last (AL) exons, Single (SES) and Multiple Exon Skipping (MES) events, Alternative splice sites (A5′SS and A3′SS), Mutually Exclusive exons (MXE), Intron Retention (RI) events in addition to alternative Transcription Start (TS) and Alternative PolyA (APA) sites. All the sequences and annotations used in this analysis were based on GRCh38 genome assembly and Gencode v27 annotation. To ensure the quantification of expressed events, a prefiltering criterion was applied by only considering those splicing events whose supporting reads were at least 10 in at least two samples per condition. In addition, a splicing event with a ΔPSI value between the silencing and control conditions less than 10% (|ΔPSI| < 0.1), or whose associated posterior probability was less than 0.90, were excluded from the downstream analysis.

### 4.8. RBP-Binding Motif Enrichment Analysis

To identify RNA-binding proteins as putative regulators of the observed changes in each splicing event identified, the sequences of the regulated exon skipping events (ES), extended by ± 200 nucleotides on both sides, were scanned for the occurrence of RBP-binding motifs. For alternative polyadenylation sites, the APA nodes reported by Whippet were sorted by coordinates and classified as proximal or distal APA sites, both of which were extended by 100 nucleotides on both sides from the APA site position. The RNA-binding motifs for 105 different splicing factors collected from the RNAcompete study [67] were used to perform binding motif enrichment analysis. For a number of RBPs, the motif from different species was confirmed, in a previous study [68], to be conserved between the human and the other species. This includes RBM47 (chicken), SF1 (Drosophila), SRP4 (Drosophila), TRA2 (Drosophila), and PCBP3 (mouse). Next, the MoSEA (Motif Scan and Enrichment Analysis) package was used to search the sequence of the splicing events for the occurrence of RBP-binding motifs [68]. The tool Find Individual Motif Occurrences (FIMO) [69] was used to scan the sequences of the events for the presence of the RBP motifs using a *p*-value < 0.001 as a cut-off. The binding motif enrichment was performed by comparing the number of occurrences of the binding motifs of the RBPs in the regulated events with that observed in a pool of 100 randomly selected sequences of the same size from equivalent regions in non-regulated events (|ΔPSI| <0.01 and *p* < 0.5). Motif enrichment was performed separately for the two directions of splicing changes (ΔPSI > 0.1 or ΔPSI < −0.1). An enrichment z-score per RNA-binding motif, the region, and direction of regulation was calculated by normalizing the observed frequency in the regulated events set with the mean and standard deviation of the 100 random control sets. The 100 random control sequences were sampled from non-regulated events for each region of regulation. An RBP was considered as enriched if associated with a z-score > 1.96. The obtained z-scores per binding motif, region, and event were then visualized using the ggplot2 Bioconductor package [62].

### 4.9. Analysis of Alternative 3′UTR Usage in TCGA Data

The analysis of alternative 3′UTR usage of breast cancer tissues from the TCGA was performed by considering the data from The Cancer 3′ UTR Atlas (TC3A) [70]. In this project, the APA usage in each tumor is quantified as the Percentage of Distal polyA site Usage Index (PDUI), which indicates a fraction of a specific polyA site used among those of a specific gene. The PDUI values of 10,267 3′UTR sites were downloaded from the project website (http://tc3a.org/) by selecting those from the BRCA cohort. Then, data of 638 luminal breast cancer tumors associated with DSCAM-AS1 FPKM expression values were extracted and considered for the analysis. To test the relationship between DSCAM-AS1 and the 3′UTR usage, a Pearson correlation analysis and a Wilcoxon rank-sum test analysis (separating tumors in two classes based on the median expression of DSCAM-AS1) was performed. Only the events characterized by a significant result in both analyses were considered significant.

### 4.10. Overlap with Public hnRNPL RNA-Binding Experiments

The overlap between the genomic regions involved in the AS events upon DSCAM-AS1 silencing and the hnRNPL RNA-binding sites was performed by considering the AS region provided by Whippet extended upstream and downstream of 200 bp. The hnRNPL-binding sites considered were retrieved from data of three different studies [14,15,54]. Specifically, the three replicates of the hnRNPL iCLIP experiment from [14] were retrieved from GEO (Gene Expression Omnibus) page GSE37562, while the RIP-Seq from [15] were retrieved from GEO page GSE72841. Finally, the HITS-CLIP (High-Throughput Sequencing of RNA isolated by CrossLinking Immunoprecipitation) data from [54] were retrieved from the Dorina v2.0 database [71], considering the results from untreated CD4+ and Jurkat cell lines. The coordinates of the hnRNPL–RNA-binding peaks were converted in hg38 using the Liftover webtool (https://genome.ucsc.edu/cgi-bin/hgLiftOver) and the overlap with the regions involved in AS events was performed using the coverageBed function of the BEDTool [72]. An overlap was considered valid if associated with a coverage greater than zero.

## 5. Conclusions

The data presented here represent a definite advancement in understanding the role of the lncRNA *DSCAM-AS1* in BC cell growth. Transcriptome analyses clearly indicate *DSCAM-AS1* involvement in the regulation of splicing and 3′-end usage, an aspect that is considered increasingly important in cancer biology, and that also correlates strongly with the observation that BC cells express an abnormal, and subtype-specific, variety of circRNAs [73]. Finally, we believe that *DSCAM-AS1* has potential as a marker of luminal BC that can be exploited for studies on the resistance to endocrine treatments in advanced BC.

## Figures and Tables

**Figure 1 cancers-12-01453-f001:**
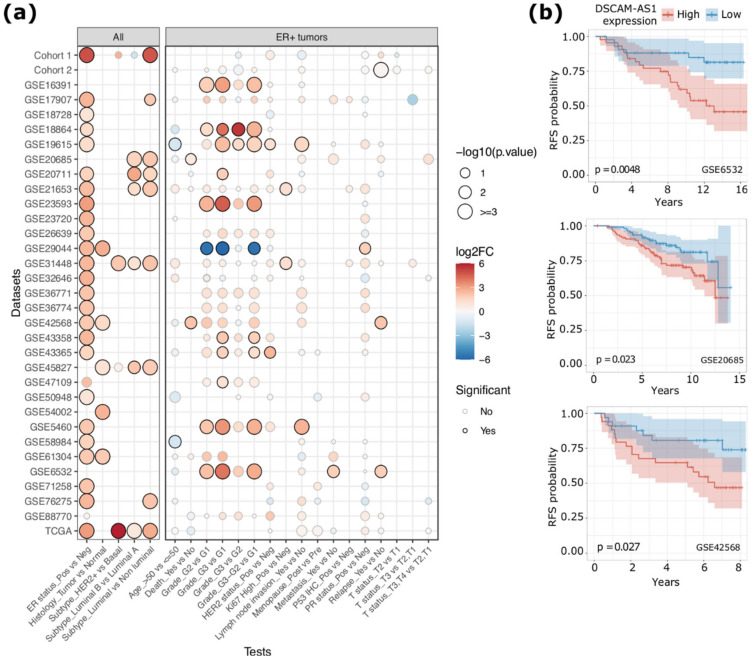
(**a**) Dot plot reporting the level of statistical significance of the differential *DSCAM-AS1* expression analyses between groups of Breast Carcinoma (BC) patients classified based on specific clinical data. The size of the dot is proportional to the significance of the results while the color code represents the log2FC of *DSCAM-AS1* expression. The left panel reports the results obtained considering all the samples, while the right panel reports the results of tests performed considering only the ER+ tumors. Estrogen Receptor (ER); positive (Pos); negative (Neg); Progesterone Receptor (PR). (**b**) Kaplan-Meier curves representing the Relapse Free Survival (RFS) of BC patients grouped by the median level of *DSCAM-AS1* expression. *p*-value by log-rank test.

**Figure 2 cancers-12-01453-f002:**
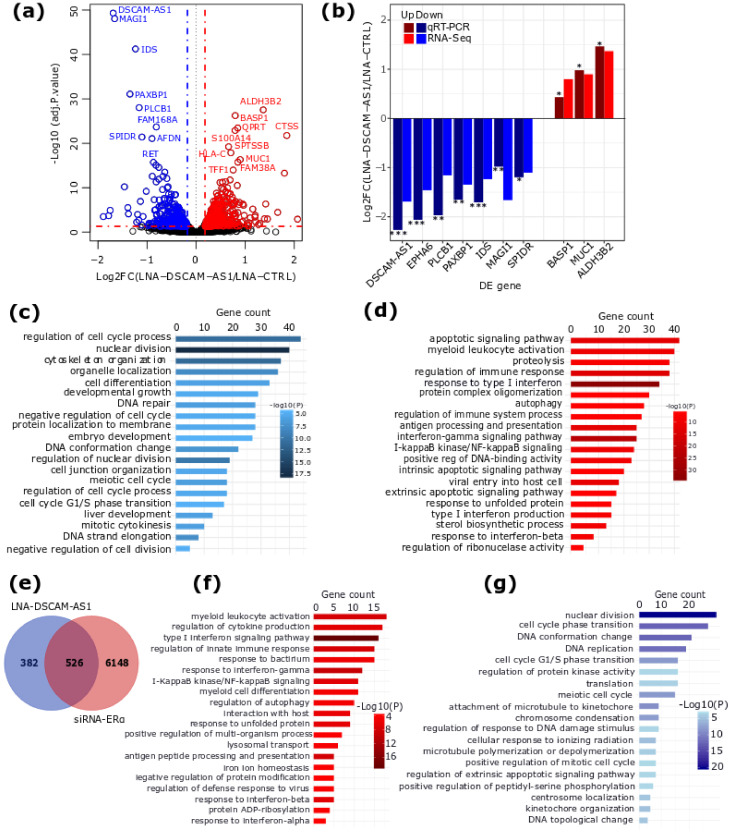
(**a**) Volcano plot showing the log2FC of gene expression and the statistical significance of the differential expression (DE) analysis performed between MCF-7 cells transfected with control or *DSCAM-AS1*-targeting LNA GapmeRs. In red are the upregulated genes, while in blue are the downregulated ones. The top first 20 significant DE genes are labeled. (**b**) Bar plot showing the expression log2FC of 10 DE genes whose expression was measured by qRT-PCR. Data from three biological replicates and *p*-value by T-test test: ***, *p*-value < 0.001; **, *p*-value < 0.01; *, *p*-value < 0.05. (**c**,**d**) Bar plot reporting the top 20 significantly enriched biological processes related to downregulated and upregulated genes, respectively. The number of genes per GO (gene ontology) term is shown and the color code is proportional to significance. (**e**) Venn Diagram showing the overlap between genes DE in this study and those DE upon ERα silencing [13]. (**f**,**g**) Top 20 enriched GO terms of overlapping DE genes showing concordant regulation are shown for genes upregulated (**f**) and downregulated (**g**) in both datasets.

**Figure 3 cancers-12-01453-f003:**
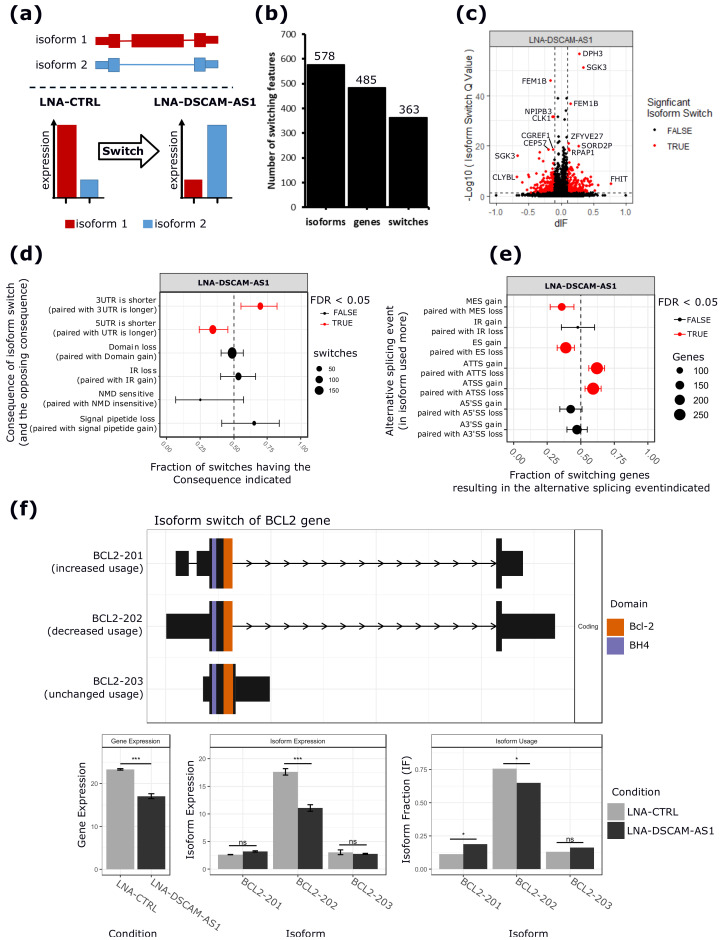
Isoform switching analysis. (**a**) Schematic overview of isoform switching concept used by the IsoformSwitchAnalyzeR tool [27]. An isoform switching event is defined as a case where the relative contribution of the isoforms to the parent gene expression changes significantly between conditions. (**b**) Bar plot reporting the number of isoforms and genes showing significant switching events upon *DSCAM-AS1* silencing. (**c**) Volcano plot showing the differential isoform fraction (dIF) and the significance of the switching isoforms. In red are reported those isoforms with a |dIF| > 10% and an FDR < 0.05. (**d**) Summary plot showing the enrichment of specific isoform features (consequence) resulting from the observed isoform switching events. From left to right, the x-axis of the plot shows the fraction of switches having the indicated consequence, where <0.5 means depleted while >0.5 means enriched upon *DSCAM-AS1* silencing. (**e**) AS event enrichment involved in isoform switches upon *DSCAM-AS1* silencing. The x-axis shows the fraction of genes showing enrichment of a specific AS event upon *DSCAM-AS1* silencing (from left to right). (**f**) Isoform switching event of the *BCL2* gene upon *DSCAM-AS1* silencing. The short 3′UTR isoform *BCL2-201* is more commonly used, while the longest 3′UTR isoform *BCL2-202* shows a decreased level upon *DSCAM-AS1* silencing. The different protein domains, *Bcl-2* and *BH-4*, of the isoforms are represented by different colors. The bar plots below show changes in the expression of the *BCL2* gene, *BCL2* isoforms, and their usage upon *DSCAM-AS1* silencing. Multiple Exon Skipping (MES); Intron Retention (IR); Exon Skipping (ES); Alternative Last (AL); Alternative First (AF); Alternative 5′ Splicing Site (A5′SS); Alternative 3′ Splicing Site (A3′SS); ***, adjusted-*p*-value < 0.001; **, adjusted-*p*-value < 0.01; *, adjusted-*p*-value < 0.05; ns, non-significant.

**Figure 4 cancers-12-01453-f004:**
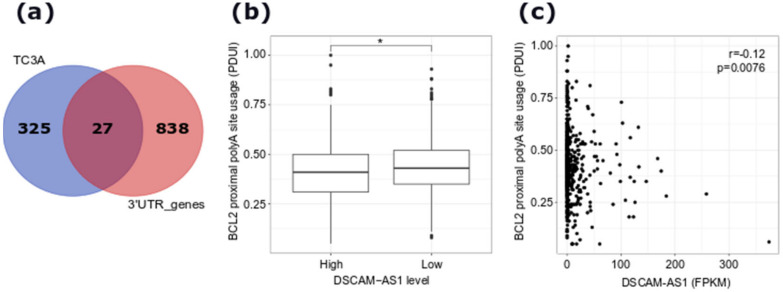
(**a**) Venn diagram showing the overlap between the genes with a differential polyA site usage and 3′UTR regulation in our dataset and those retrieved from The Cancer 3′ UTR Atlas (TC3A) database with differential 3′UTR usage showing correlation with *DSCAM-AS1* expression in BCs. (**b**) Boxplot showing the level of BCL2 proximal polyA site usage from TC3A database data considering luminal BCs characterized by a high or a low DSCAM-AS1 expression; *, Wilcoxon rank-sum test *p*-value < 0.05. (**c**) Scatterplot showing the relationship between BCL2 proximal polyA site usage index and DSCAM-AS1 expression (in FPKM) in BC samples retrieved from the TC3A database. Proximal to Distal polyA site Usage Index (PDUI).

**Figure 5 cancers-12-01453-f005:**
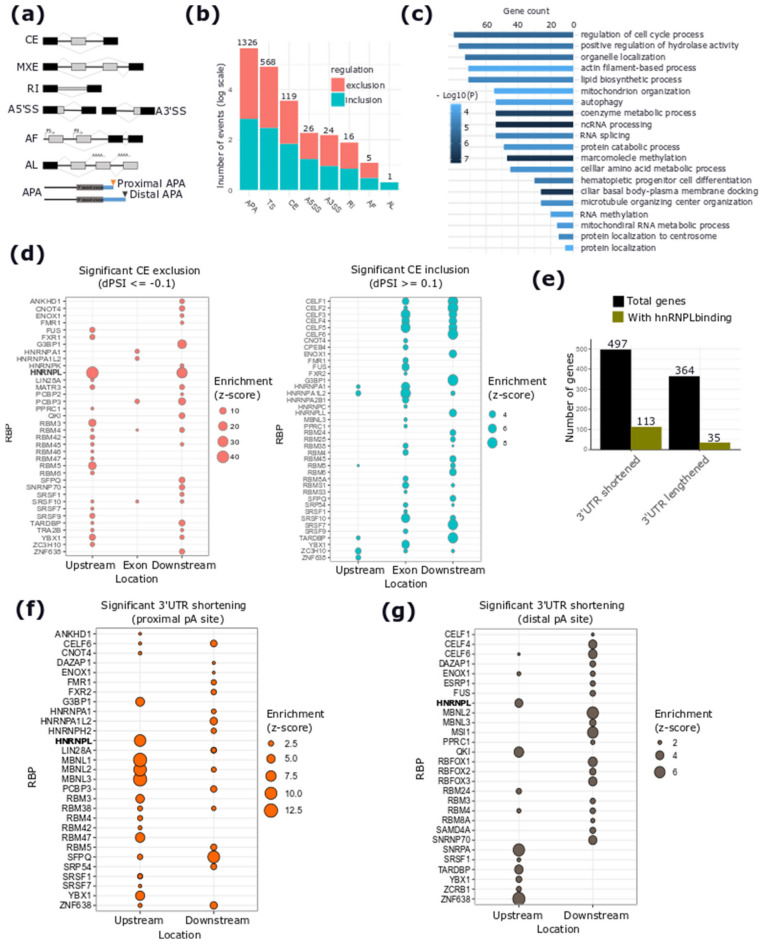
Alternative splicing (AS) changes upon DSCAM-AS1 silencing and prediction of binding motifs for RNA-binding proteins (RBPs). (**a**) Schematic depiction of the different classes of AS events analyzed using Whippet. Exon Skipping (ES); Mutually Exclusive Exons (MXE); Intron Retention (IR); Alternative 5′ (left) and 3′ (right) Splice Sites (ASS); Alternative First Exons (AF); Alternative Last Exons (AL); Alternative PolyA (APA) site; tandem Transcriptional Start Sites (TS). (**b**) Number of significant AS events per type and direction of regulation (inclusion or exclusion). (**c**) GO-enriched terms related to genes showing significant AS events. (**d**) RBP-binding motif enrichment results for significant CE (cassette exon) exclusion (left panel) and inclusion events (right panel), significance: z-score > 1.96. (**e**) Number of genes with significant 3′UTR splicing (shortening and lengthening) with indication of the number of those having a predicted heterogeneous nuclear ribonucleoprotein L (hnRNPL)-binding motif. (**f**,**g**), RBP-binding motif enrichment results for genes with significant 3′UTR shortening upon *DSCAM-AS1* silencing, considering a region upstream and downstream of the proximal (**f**) or the distal (**g**) APA sites, respectively, significance: z-score >1.96.

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
