# Peer review of "DSCAM-AS1-Driven Proliferation of Breast Cancer Cells Involves Regulation of Alternative Exon Splicing and 3′-End Usage"

_cancers, 2020, doi:10.3390/cancers12061453_

Round 1
Reviewer 1 Report
The article: DSCAM-AS1-driven proliferation of breast cancer cells involves regulation of alternative exon splicing and 3´- end usage, summarizes the results of basic biomedical research focused on the breast cancer mainly. The article is original and offers scientifically valuable and interesting results – showing the expression specificity of lnRNA DSCAM-AS1 in relation to particular breast cancer subtypes. Interestingly, the results show the influence of DSCAM-AS1 expression level on splicing giving the potential clue for explanation of cancer-specific splicing phenomenon. This is novel and relevant result! However, there are several serious shortcomings within the experimental design and used methodology that disable the article to be published in its current form. Therefore, I suggest to reconsider the acceptance after major revision of following issues:
- The RNA isolated from “cohort” samples needs to be qualitatively characterized prior to any conclusive RNA seq experiments! Supplements the results by analysing RNA integrity number of used samples.
- The results 2.1. indicate that the difference of the DSCAM-AS1 expression level between ER+ and ER- is statistically significant in 28 out of 30 analysed microarray datasets. However, on the figure 1.A. there are 23 significantly different datasets! Is there any reason for that?
- The results 2.1. indicate association of DSCAM-AS1 expression with relapse rate in 3 out of 10 microarray datasets. What is the association in remaining datasests?
- The DSCAM-AS1 has been described here as lnRNA overexpressed in subset of breast tumours, however all the in vitro experiments operates with down-regulation only. Is there any reason for that? Is DSCAM-AS1 overexpressed in MCF-7 cells in comparison with analysed breast cancer samples?
- The gene expression is complex process that react very sensitively to various stimuli – including the chemical transfection. To distinguish the DSCAM-AS1 related expression changes from those caused by transfection itself the result should be compared with non-transfected model cells too. Please supplement the results.
- The knock-down experiments is not properly described in method section. It is unclear what cell population was analysed after transfection of LNA, what is the transfection efficiency, how many biological and technical replicates were analysed and what represent the result – mean of particular results? How was done the normalisation of expression of analysed genes in case of usage of mixed population after transient transfection (transfected and non-transfected cells) discussed in results 2.2.?
- The specificity of DSCAM-AS1 downregulation-mediated genes expression and splicing profile is not proved by rescue experiment.
- Though MCF-7 are generally considered as ER+ it would be worthy to prove its ER status.
- In method section are mentioned SK-BR-3 cells, however no experimental results are related to this cell line. Why? Generally, the in vitro experiments done on single cell line are not very conclusive!
- How do you explain the fact that HNRNPL expression is unaffected by DSCAM-AS1 even though they interact with each other?
Author Response
Please see the attachment: Rebuttal_letter_to_Reviewer_1

Reviewer 2 Report
DSCAM-AS1 is a long noncoding RNA with oncogenic power in several tissues and in breast cancer (BC) where its expression is dependent on Estrogen Receptor Alpha (ERα)
Using several publically available RNA-seq and microarray datasets and RT-PCR validation on 2 cohorts from primary cancer tissues, the authors confirmed the relation between the DSCAM-AS1 expression and the poor survival rate of BC patients. Then they investigate the functionality of this lncRNA by performing LNA-gapmer mediated knock-down in MCF7 cells following by a RNA-seq transcriptomic analysis. This approach confirm a previous publication obtained by siRNA-mediated silencing of this lncRNA.
Analysis of alternative isoform expression show also that DSCAM-AS1 may be involved in the alternative splicing and processing of RNA. Because it has been reported that hnRNPL interacts with this lncRNA (confirmed here by CLIP assay), the authors investigate whether alternative splicing perturbation by DSCAM-AS1 may be mediated by hnRNPL. Only the cassette exon skipped upon DSCAM-AS1 KD, were found enriched for binding site of hnRNPL, whereas included variant exons are predicted to be bound by other RBPs.
Altogether, these data illustrate the potential functional role of DSCAM-AS1 in regulating isoforms expression.
This study is well conducted with good controls where it is required. The manuscript is well written. Several data of this study confirm and extent already published oncogenic role of DSCAM-AS1, that is very interesting. The other interest of this study is the description of the functional role of DSCAM-AS1 depending of the hnRNPL, a splicing factor that interacts with this lncRNA. However, it is not clear how DSCAM-AS1 could contribute directly to the hnRNPL function since the argument is mainly a study of hnRNPL binding site enrichment on DSCAM-AS1 dependent variant exon. Indeed, The strength of these data on the alternative splicing regulation should be increase to sustain better the conclusion.
First, some alternative splicing event such as for Blc2, should be validated by an other approach than bioinformatic analysis, such as RT-PCR assay on RNA from the cohort or from MCF7 depleted of DSCAM-AS1. Other examples (as the CCNE1 variation on the 3'UTR) should be validated by RT-PCR.
Second, at least by bioinformatic analysis, the following question should be addressed : are the DSCAM-AS1 dependent genes for their alternative splicing also found in the lists of genes known to be alternatively spliced by hnRNPL?
Among genes which are expressed in both cell types could we found common variant exons for hnRNPL and DSCAM-AS1 ?
Hung et al. (Bindereif team) in RNA. 2008 Feb;14(2):284-96. Epub 2007 Dec 11.
Diverse roles of hnRNP L in mammalian mRNA processing: a combined microarray and RNAi analysis. [GSE8945].
Or
how many of the skipped exon in a DSCAM-AS1 dependent manner have been identified to be a target of hnRNPL by iCLIP. (REF#14, Rossbach et al. 2014 RNA biol.)
Third, I think it is never clearly write what is the effect of DSCAM-AS1 expression on hnRNPL function. Does it lead to a repressive effect of hnRNPL and regulation induced by DSCAM-AS1 KD should be opposite to hnRNPL KD ; or does it lead to enhance the hnRNPL effect, and both effect could be additive ?
Since only regions of the cassette exon skipped upon DSCAM-AS1 KD are enriched for hnRNPL binding sites, it could be extrapolated that skipping of these cassette exons appears when hnRNPL is becoming free of the lncRNA. This mean that hnRNPL is mainly a negative regulator of cassette exon inclusion. Does it correspond to the litterature ? this point should be discussed.
minor points :
Since "different subset of RBPs was identified to be enriched for exons showing more inclusion levels upon DSCAM-AS1 silencing (Figure 4d,right panel)", did these RBPs were found upregulated or differentially expressed or spliced after the knock-down of DSCAM-AS1 ? Did the differentially expression of RBPs can explain the observed alternative splicing?
Among the RBPs identified through their enrichment binding sites in DSCAM-AS1 dependent variant exons, are there some that interact with DSCAM-AS1, as it has been described for hnRNPL ? How DSCAM-AS1 is predicted to bound these RBPs ? can one of this prediction can be validated by CLIP or in litterature ? This point should be adressed or at least discussed.
The following sentence line 247 "Interestingly, the BCL2 gene, whose RNA stability was previously demonstrated to be tightly regulated by hnRNPL [16],"
should be town-down because at least a contradictory study has investigated the question and didn't have the same conclusion.
Effect of Modulation of hnRNP L Levels on the Decay of bcl-2 mRNA in MCF-7 Cells
Mi-Hyun Lim 1 , Dong-Hyoung Lee, Seung Eun Jung, Dong-Ye Youn, Chan Sun Park, Jeong-Hwa Lee
. 2010 Feb;14(1):15-20. Korean J Physiol Pharmacol
doi: 10.4196/kjpp.2010.14.1.15. PMID: 20221275
The suplementary figure7 showing the hnRNPL levels of protein after KD of DSCAM-AS1 is not convincing because the revelation / picture is saturated and even this, a sligth decrease is observable, and not for GAPDH. The "uncropped WB images" showing all the bands considered for the quantification should be labelled to indicate which lane is control and which one is KD. Does the RNA levels of hnRNPL is also 'unchanged" after the DSCAM-AS1 KD ?
In fact, the protein level has not to be unchanged after DSCAM-AS1 KD because the trapping effect of DSCAM-AS1 can be accompagnied by hnRNPL protein stabilization, but it is interesting to have a clear result of this assay.
Author Response
Please see the attachment: Rebuttal_letter_to_Reviewer_2

Round 2
Reviewer 1 Report
I would like to thank the authors for clarification of my doubts. The explanations proved high scientific level and good orientation of authors in problematic discussed in the article. The article in the modified form meets the criteria for admission to the journal. Therefor I recommend it for acceptation in the current form.
Author Response
No action required
Reviewer 2 Report
Point1
I agree that containment makes it difficult to re-experiment. Nevertheless, it should be negotiable with the editor, since everyone is affected by this pandemia.
That the authors have tried to validate the splicing change of Blc2 observed in MCF7 RNA-seq in bioinformatics studies of other cancer datasets is a valid approach.
Point2
Ok
Point3
ok
I agree with the reviewer 1 that RNA integrity analysis is fundamental to interpreting the quantification of the RNA level in the analysis of the cohorts (figure1). I am sorry that I am not at all convinced by the authors' response, suggesting that their data by RT-qPCR might be less sensitive to RNA degradation than when doing RNA-seq. Even in RT-qPCR, if the amplicon is always 99 bp regardless of the samples, this does not mean that the RT-qPCR measurements are quantitatively reliable, as there may be an (unknown) proportion of RNA that cannot be amplified by PCR since it could be degraded.
At the very least, the authors could add is the supplementary table 1b/1c the Ct obtained for the 18S (line 507) quantified in all the cohort samples in order to demonstrate that the ribosomal RNAs are not too degraded for a constant quantity of RNA put in the reverse transcription assay. They could also sort the samples according to RNA quality, and ensure that they show the same trend with only the higher quality RNAs. Another possibility would have been to confirm their first normalized quantification with the 18 S (line 507), with another normalizer gene that expresses at about the same level as DSCAM-aS1 to remove the bias of the too different expression levels between normalizer and gene of interest.
Author Response
Please see the attachment "Rebuttal_2"
